# Immune Profile of BRAF-Mutated Metastatic Colorectal Tumors with Good Prognosis after Palliative Chemotherapy

**DOI:** 10.3390/cancers14102383

**Published:** 2022-05-12

**Authors:** Jeong Eun Kim, Ji-Hun Kim, Sang-Yeob Kim, Hyungwoo Cho, Yeon-Mi Ryu, Yong Sang Hong, Sun Young Kim, Tae Won Kim

**Affiliations:** 1Department of Oncology, Asan Institute for Life Sciences, Asan Medical Center, University of Ulsan College of Medicine, Seoul 05505, Korea; jeongeunkim@amc.seoul.kr (J.E.K.); hwcho@amc.seoul.kr (H.C.); yshong@amc.seoul.kr (Y.S.H.); sunyoungkim@amc.seoul.kr (S.Y.K.); 2Department of Pathology, Asan Institute for Life Sciences, Asan Medical Center, University of Ulsan College of Medicine, Seoul 05505, Korea; jihunkim@amc.seoul.kr; 3Department of Optical Imaging Core, Asan Institute for Life Sciences, Asan Medical Center, University of Ulsan College of Medicine, Seoul 05505, Korea; sykim3yk@amc.seoul.kr (S.-Y.K.); ryu4177@amc.seoul.kr (Y.-M.R.)

**Keywords:** CRC, BRAF, prognosis, immune profile

## Abstract

**Simple Summary:**

The present study demonstrated that the distinct subgroup of BRAF-MT CRC showed a good response after palliative chemotherapy. Based on the immune profile analysis, higher PD-L1 expression and CD8-positive cell infiltration were shown in this study population. Furthermore, the assessment of the immune profile of BRAF-MT tumors can be helpful to better understand tumor biology and the different clinical outcomes of BRAF-MT CRC.

**Abstract:**

Background: BRAF-mutated colorectal cancers (BRAF-MT CRCs) are known to have poor prognoses. BRAF-MT CRC was reported to be possibly related to the immune-activated phenotype. Objectives: This study aimed to investigate the association between the immune microenvironment and prognosis of BRAF-MT CRC. Methods: We evaluated clinical outcomes and investigated the immune profile of the BRAF-MT CRC tumors using the multiplex immunohistochemistry of immune-related markers: cytokeratin, programmed death ligand-1 (PD-L1), programmed cell death protein-1 (PD-1), and a cluster of differentiation 8 (CD8). Results: Out of 2313 tumors, 123 were BRAF-MT tumors. Among them, 86 tumors with available tissue were included. Out of 86 patients, 75 patients were non-good responders (GR), whereas 11 patients were GR. Median progression-free survival after first-line chemotherapy (4.6 vs. 12.4 months, *p* = 0.008) and overall survival (11.8 vs. 45.0 months) were longer in the GR group (*p* < 0.001). Median CD8+ T cell (254.29 vs. 656.0, *p* = 0.092), PD-L1+ tumor cell (0.95 vs.15.56, *p* = 0.050), PD-L1+ stromal cell (3.17 vs. 72.38, *p* = 0.025), PD-L1+ tumor and stromal cell (5.08 vs. 74.92, *p* = 0.032), and PD-1+ stromal cell (45.08 vs. 325.40, *p* = 0.046) counts were greater in the GR group. Conclusion: The clinical outcomes of unselected patients with BRAF-MT CRC were generally similar to those in previous studies. Based on the immune profile analysis, higher PD-L1 expression and CD8-positive cell infiltration were observed in BRAF-MT tumors with a good prognosis.

## 1. Introduction

Colorectal cancer (CRC) is the major leading cause of death and the most commonly diagnosed gastrointestinal cancer [1,2] Diagnosis with molecular subtyping, including testing for RAS (rat sarcoma) and rapid accelerated fibrosarcoma mutations, has become a standard approach to the initial work-up of patients with metastatic colorectal cancer (mCRC). Approximately 50% of CRC patients have RAS mutation, and 10% of CRC cases have BRAF mutation. The V600E mutation in BRAF is the most common BRAF mutation and is generally associated with a poor prognosis due to an aggressive phenotype; the higher the rate of peritoneal and metastatic lymph node involvement, the lesser the efficacy of standard cytotoxic combination chemotherapy [3,4,5].

Targeted therapy for BRAF-mutant (BRAF-MT) mCRC was prosed based on the successful outcomes of BRAF inhibition in melanoma. However, BRAF inhibitor monotherapy showed a low response rate in BRAF-MT mCRC [6,7,8] because rebound feedback with increased signaling through the epidermal growth factor receptor (EGFR) pathway after BRAF inhibition occurs in colon cancer [9,10]. Recently, BRAF inhibitors combined with anti-EGFR monoclonal antibodies were proven to be effective. Combination treatment with dabrafenib, trametinib, and panitumumab showed clinical activity in BRAF-mutated colorectal cancer (BRAF-MT CRC) in a single-arm phase II study of [11]. The addition of vemurafenib to irinotecan and cetuximab improved progression-free survival (PFS) and tumor response in BRAF-MT CRC that was previously treated with one or two regimens of palliative chemotherapy in another prospective randomized phase II study [12,13]. Combination treatment with encorafenib and cetuximab, with or without binimetinib, showed significantly longer overall survival (OS) with a higher response rate than standard therapy in BRAF-MT CRC recently [14,15,16].

The unique biology related to poor outcomes of BRAF V600E mutation (BRAF-MT^V600E^) does not apply to non-V600 mutations in BRAF [17]. Moreover, in the BRAF-MT^V600E^ tumors, heterogeneity has been evaluated by gene expression signatures. These tumors can be divided into two subtypes, BM1 and BM2, by the expression profiles independent of MSI status, PI3K mutation, gender, and sidedness. BM1 is characterized by KRAS/AKT pathway activation and epithelial–mesenchymal transition (EMT) and stronger treatment effect after combination treatment with dabrafenib, trametinib, and panitumumab was observed in this subtype [18]. On the other hand, BM2 shows the important deregulation of the cell cycle [19].

Approximately half of BRAF-MT CRC cases are classified as consensus molecular subtype 1 (CMS1), and the proportion of CMS1 is high even in microsatellite stable patients. These findings reflect an increased recognition of the immune-activated phenotype of BRAF-MT CRC [20]. In another study, programmed cell death ligand-1 (PD-L1) expression was shown to be related to BRAF mutation [21]. In a recent retrospective analysis, BRAF mutation was suggested to be a potential predictive biomarker of immune checkpoint inhibitors in mismatch repair (MMR)-deficient CRC with a lower response rate and shorter PFS [22].

Therefore, we evaluated the immune profile of the BRAF-MT CRC tumors and compared the differences in the immune profile of the tumors according to treatment response to investigate the association between immune microenvironment and prognosis of BRAF-MT CRC.

## 2. Materials and Methods

### 2.1. Patients and Data Collection

Between January 2005 and October 2015, the data of patients with metastatic or recurrent colorectal cancer who underwent *BRAF* mutation testing at the Asan Medical Center were reviewed. The clinicopathological characteristics and survival of both groups were analyzed. Clinical data, including age at diagnosis, sex, tumor location, initial stage, and treatment, were retrieved from the patient medical records. Right-sided cancers included tumors from the cecum to the transverse colon, whereas left-sided cancers included tumors from the splenic flexure to the rectosigmoid junction. This study was approved by the Institutional Review Board of the Asan Medical Center and conducted in accordance with the principles of the Declaration of Helsinki. The IRB waived the requirement for informed consent for this retrospective study (2011-0511).

In this study, patients with BRAF-MT tumors were included and divided into two groups according to treatment response after palliative chemotherapy: the good responder (GR) group and the non-good responder (non-GR) group. GR was defined as patients with sustained partial response at least 6 months after the first-line palliative chemotherapy, which was longer than the median time-to-progression, to palliative first-line chemotherapy of the study population [23].

### 2.2. Tumor Tissue Sampling and Mutational Analysis

For genetic analyses, formalin-fixed paraffin-embedded (FFPE) tissues of primary or metastatic lesions were used. A pathologist reviewed the FFPE tissue blocks retrieved from the archives and marked the tumor portion. After cutting tissue blocks, the genomic DNA was extracted. Direct sequencing or real-time PCR were used to test for the *BRAF* (exon 15) mutation. The MMR status was determined using PCR or immunohistochemistry (IHC).

### 2.3. Immune Profile Using Immunofluorescence Staining

Immune profile was evaluated in patients with BRAF-MT CRC with available tumor tissue. From each case, one matched FFPE tissue block from the selected H&E pretreatment biopsy slide was obtained. Four-micron-thick slices were cut from the tissue and transferred onto plus-charged slides. After, multiplexed fluorescent IHC was performed using the Leica Bond Rx™ Automated Stainer (Leica Biosystems, Nussloch GmbH, Nussloch, Germany). The slides were baked at 60 °C for 40 min, and deparaffinization was performed using Leica Bond Dewax solution (Cat #AR9222; Leica Biosystems). Antigen retrieval with Bond Epitope Retrieval 2 (Cat #AR9640; Leica Biosystems) for 30 min was performed afterwards. Next, the slides were incubated with primary antibodies as well as secondary horseradish peroxidase-conjugated polymers. Each horseradish peroxidase-conjugated polymer created a covalent binding of a different fluorophore using tyramide signal amplification. This covalent bond reaction was followed by additional antigen retrieval with Bond Epitope Retrieval 1 (Cat #AR9961; Leica Biosystems, Milton Keynes, UK) for 20 min to remove prior primary and secondary antibodies before the next step in the sequence. Four sequential rounds of staining were performed for each slide. After sequential reactions, the sections were counterstained with Spectral DAPI and mounted with HIGHDEF^®^ IHC fluoromount (Enzo Life Sciences, Farmingdale, NY, USA).

The sections were stained using an Opal™ 7-color Automation IHC Kit (Perkin Elmer, Waltham, MA, USA). The cells were stained with antibodies against cytokeratin (CK, 1:300; Novus, CO, USA), cluster of differentiation 8 (CD8, 1:300; Bio-RAD, Hercules, CA, USA), programmed death-ligand 1 (PD-L1, 1:500; CST), and programmed cell death protein-1 (PD-1, 1:300; Abcam, Cambridge, UK), and the fluorescence signals were captured with the following fluorophores: Opal 520, Opal 570, Opal 650, and Opal 690.

### 2.4. Image Acquisition and Quantitative Data Analysis

The Vectra^®^ 3.0 Automated Quantitative Pathology Imaging System (Akoya Biosciences, Marlborough, MA/Menlo Park, CA, USA), was used for the scanning of multiplex-stained slides. The images were visualized in the Phenochart whole slide viewer (Akoya Biosciences, Marlborough, MA/Menlo Park, CA, USA).

The inform 2.3 Image Analysis Software (Akoya Biosciences, Marlborough, MA/Menlo Park, CA, USA) and Spotfire software (TIBCO Software Inc., Palo Alto, CA, USA) were used for analyzing images. The threshold for the positivity of each marker is determined by the pathologist using IHC scoring: >2.8 for CK, >14 for CD8, >10 for PD-L1, and >1.5 for PD-1. The percentage of each immune cell subset was calculated by dividing the absolute number of each subset by the total number of all the cells.

### 2.5. Statistical Analysis

PFS was defined as the time between the start date of chemotherapy and the date of the documented disease progression or death, whichever occurred first. OS was defined as the time from the start date of chemotherapy to the date of death from any cause. Survival rates and corresponding standard errors were estimated using the Kaplan–Meier method. Survival curves were compared by the log-rank test. The Pearson’s chi-square test or Fisher’s exact test for the categorical variables and Student’s *t*-test or Mann–Whitney U test for the continuous variables were used to compare baseline characteristics of the groups. A two-sided *p*-value of <0.05 was considered statistically significant. All statistical calculations were conducted using R version 3.6.2. (R Foundation for Statistical Computing, Vienna, Austria, https://www.R-project.org/, accessed on 15 September 2021).

## 3. Results

### 3.1. Patient Characteristics

A total of 2311 patients with metastatic or recurred colorectal cancers underwent BRAF testing. Among them, 123 patients who had BRAF-mutated CRC were included in the present study; 119 had a BRAF mutation of V600E, 3 had a BRAF mutation of K6001E, and 1 had a BRAF mutation of V600E and K600E. A total of 104 patients were included for further analysis, except for 19 patients without available tissue. The pathologist reviewed 104 tumors with BRAF mutation and analyzed the immune-related pathologic variables using 98 tumor tissues, except for 6 tumors unavailable for multiplex IHC. Six patients who underwent surgery with curative intent and six patients who were lost to follow-up before palliative chemotherapy were excluded from our analysis. A total of 86 tumors were included in the final analysis (Figure 1). Of the 86 patients, 75 patients were non-GR, whereas 11 patients were GR. The baseline characteristics, including sex, sidedness, tumor grade, histology, stage at diagnosis, KRAS/NRAS mutation status, and MSI status, were not different between the two groups (Table 1).

The majority of the patients received FOLFIRI or FOLFOX/XELOX chemotherapy with or without targeted agents as first-line palliative chemotherapy in both groups. One patient in the non-GR group received combination treatment with cetuximab, BRAF inhibitors (LGX818), and PI3K inhibitors (BYL719). Three patients in the non-GR group could not receive palliative chemotherapy due to poor performance status and/or rapid progression of the tumor just before the planned infusion of chemotherapy.

### 3.2. Survival Outcome of Patients with BRAF-MT Tumors

While the median follow-up duration was 32.1 months (95% confidence interval (CI): 19.4–61.7), the median PFS after first-line chemotherapy (PFS1) was 5.28 months (95% CI: 4.3–7.5, Figure 2), and the median OS was 13.7 months (95% CI 10.3–17.9, Figure 2) in 86 patients included for the final analysis, respectively. The median PFS after second-line chemotherapy (PFS2) was 2.62 months (95% CI: 2.0–4.3, Figure 2). Moreover, the median PFS1 was 4.6 months (95% CI: 4.1–5.7) in the non-GR group vs. 12.4 months (95% CI: 11.5–not available (NA)) in the GR group (*p* = 0.008) (Figure 3A). The median OS was 11.8 months (95% CI: 9.8–15.8) in the non-GR group vs. 45.0 months (95% CI: 27.7–NA) in the GR group (*p* < 0.001) (Figure 3B). The median PFS2 was 2.3 months (95% CI: 1.9–4.0) in the non-GR group vs. 5.5 months (95% CI: 4.3–NA) in the GR group (*p* = 0.021) (Figure 3C).

### 3.3. Immune Profile of the BRAF-MT Tumors

The immune profiles of the BRAF-MT tumors were compared according to the groups: non-GR and GR groups (Table 2). The median CD8+ T cell count was greater in the GR group than in the non-GR group without statistical significance (Figure 4A): 254.29 (interquartile range (IQR): 79.21–580.43) in the non-GR group vs. 656.0 (IQR: 149.05–1067.14) in the GR group (*p* = 0.092). The median PD-L1+ tumor cell count was significantly greater in the GR group (Figure 4B): 0.95 (IQR: 0.00–7.32) in the non-GR group and 15.56 (IQR: 2.13–73.81) in the GR group (*p* = 0.050). The median PD-L1+ stromal cell count was also significantly greater in the GR group (Figure 4C): 3.17 (IQR: 0.00–27.30) in the non-GR group and 72.38 (IQR: 3.65–178.25) in the GR group (*p* = 0.025). Moreover, the PD-L1+ tumor and stromal cell counts were significantly greater in the GR group (Figure 4D): 5.08 (IQR: 0.36–49.68) in the non-GR group and 74.92 (IQR: 12.38–243.49) in the GR group (*p* = 0.032). PD-1+ stromal cell count was also significantly greater in the GR group: 45.08 (IQR: 19.39–174.66) in the non-GR group and 325.40 (IQR: 69.52–423.87) in the GR group (*p* = 0.046).

## 4. Discussion

To the best of our knowledge, this is the first study to evaluate the differences in the immune profile of BRAF-MT CRC according to treatment response to investigate the association between immune microenvironment and the prognosis of BRAF-MT CRC. The clinical outcomes of the unselected patients with BRAF-MT CRC treated in a real clinical setting were generally similar to those in previous studies. However, PFS and OS in a distinct subgroup of BRAF-MT CRC showed GR after palliative chemotherapy was better than the non-GR group and comparable to BRAF-WT CRC. Based on the immune profile analysis, the PD-L1+, CD8+, and PD-1+ stromal cell counts were shown to be higher in the tumors of the GR group. These results suggest that a better response to palliative chemotherapy and the survival outcomes of the BRAF-MT tumors might be associated with the different immune microenvironments of the tumors.

In the consensus molecular subtypes (CMS1–CMS4), the BRAF mutations tend to be more prevalent in CMS1 CRC. However, BRAF mutation can also be found in CMS3 and CMS4 CRC. According to the transcriptional subtypes of BRAF-MT^V600E^ CRC, the BM1 subtype was associated with a poorer prognosis, which is characterized by KRAS/AKT pathway activation, EMT that mediates chemotherapy resistance, and increased immune reactivity, whereas BM2 shows the deregulation of the cell-cycle checkpoints and was associated with better prognosis [18,24]. Therefore, in addition to the known mutation in CRC, BRAF mutation, transcription, and immunologic features should be evaluated to understand tumor biology and therapeutic vulnerability.

The prognostic value of PD-L1 expression has not been clearly determined. Some studies indicated that PD-L1 expression on tumor cells was a negative prognostic marker related to poor survival outcomes [25,26,27]. However, in other studies, PD-L1 expression was not related to prognosis [28,29]. These conflicting results can be explained by a non-standardized methodology evaluating PD-L1 expression, tumor heterogeneity, various patient populations, and the complex interaction between PD-L1 and other immune markers [30,31]. A recent study showed that tumors with increased PD-L1 and CD8 expression were associated with better survival based on the evaluation of the tumor immune microenvironment of CRC using the IHC data of PD-L1, PD-1, and CD8 [31]. In accordance with these efforts, our study suggested the association between the immune microenvironment of BRAF-MT tumors and survival after conventional chemotherapy. Furthermore, our study investigated the difference in immune profile using multiplex IHC data, which is an approach to evaluate multiple immune cells within tumors and the tumor microenvironment simultaneously. We compared the immune profiles of the BRAF-MT tumors according to treatment response using the quantitative analysis of the immune cells.

According to the results of a recent landmark phase III trial, the combination treatment with cetuximab and encorafenib is suggested as the best treatment option for BRAF-MT^V600E^ CRC [15]. However, in MSI BRAF-MT CRC, immune checkpoint inhibitors can be a better treatment option based on the promising results of the KEYNOTE-177 trial, showing encouraging treatment responses and survival compared with the BRAF inhibitors in a refractory setting [32,33,34]. BRAF-MT CRC has more stromal cells, more immune cell infiltration, and lower tumor purity in a recent study that evaluated the immune microenvironment of BRAF-MT CRC using The Cancer Genome Atlas and Gene Expression Omnibus data. BRAF-MT CRC had the higher expression of many immunotherapeutic targets, such as PD-1, PD-L1, CTLA-5, LAG-C, and TIM-3 [35]. Our study also supports such data with higher PD-L1+, CD8+ cell, and PD-1+ stromal cell counts in the GR group irrespective of the MSI status. Therefore, immunotherapy can be suggested as a treatment option for BRAF-MT CRC based on the perspective of immune biology in addition to the MSI subtype. Even with the improvement of survival with the new treatment strategy in BRAF-MT CRC, there are still unmet needs because of the relatively short response duration of a combination treatment of encorafenib and cetuximab. Additionally, the priority or sequence of this new treatment and immunotherapy is still under debate, especially in MSI BRAF-MT CRC. With our study, we can suggest a clue to a combination of immunotherapy with the selection of patients based on the immune profile of BRAF-MT tumors. The immune profile of BRAF-MT CRC can be evaluated in future clinical trials or in the prospective cohort to enhance the understating of the tumor biology of BRAF-MT tumors and find a new treatment strategy for BRAF-MT CRC.

However, this study has several limitations. Because of the small proportion of BRAT-MT tumors in metastatic CRC and tissue availability, only a small number of patients were included in the final immune profile analysis. Moreover, even with the use of the multiplex IHC method, we could not comprehensively evaluate the tumor microenvironment. More comprehensive analysis with additional CD3+ T cell, regulatory T cell, B cell, and tumor macrophage data can be helpful to better understand the tumor microenvironment of BRAF-MT CRC. Nevertheless, this study introduces a new approach to evaluating the immune profile of BRAF-MT tumors to assess the difference in BRAF-MT CRC according to treatment response and survival. In addition to the MSI status, the immune profile might be an important factor in explaining the diverse tumor biology of BRAF-MT tumors, and the incidence of BRAF mutation was lower, with 5.3% compared to previous reports. This might be explained by possible different prevalences according to race or ethnicity. The prevalence of BRAF mutation was reported as 5.6% in Asian patients in another study. Because of the retrospective nature of data collection of our study, the prevalence of BRAF mutation might be under-investigated with the exception of patients without results of BRAF testing. In addition, only four markers were selected to evaluate the immune profile of BRAF-MT tumors due to a limitation in the number of markers for multiple IHC analysis at that time when this study was performed at our center. It was not sufficient to comprehensively assess the immune profile, but we cautiously selected well-known representative immune markers. Furthermore, there is a possibility that we cannot clearly explain the association between immune profile and the effect of chemotherapy considering heterogeneous chemotherapy regimens. However, we can suggest a clue that the de novo difference in the immune profile of BRAF-MT CRC might be associated with treatment responses considering that the tumor tissue used for the multiplex IHC to evaluate the tumor immune profile was obtained before treatment. However, these analyses should be also conducted in patients with BRAF-WT CRC to explain that these features in tumor biology are confined to BRAF-MT tumors. We additionally discussed this limitation in the discussion section.

## 5. Conclusions

In conclusion, the present study demonstrated that the distinct subgroup of BRAF-MT CRC showed a good response after palliative chemotherapy. Based on the immune profile analysis, higher PD-L1 expression and CD8-positive cell infiltration were shown in this study population. Furthermore, the assessment of the immune profile of BRAF-MT tumors can be helpful to better understand tumor biology and the different clinical outcomes of BRAF-MT CRC.

## Figures and Tables

**Figure 1 cancers-14-02383-f001:**
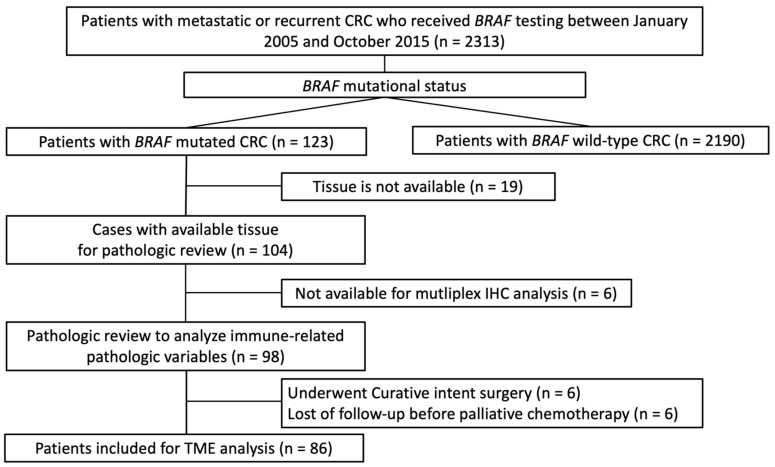
Flowchart of patient selection.

**Figure 2 cancers-14-02383-f002:**
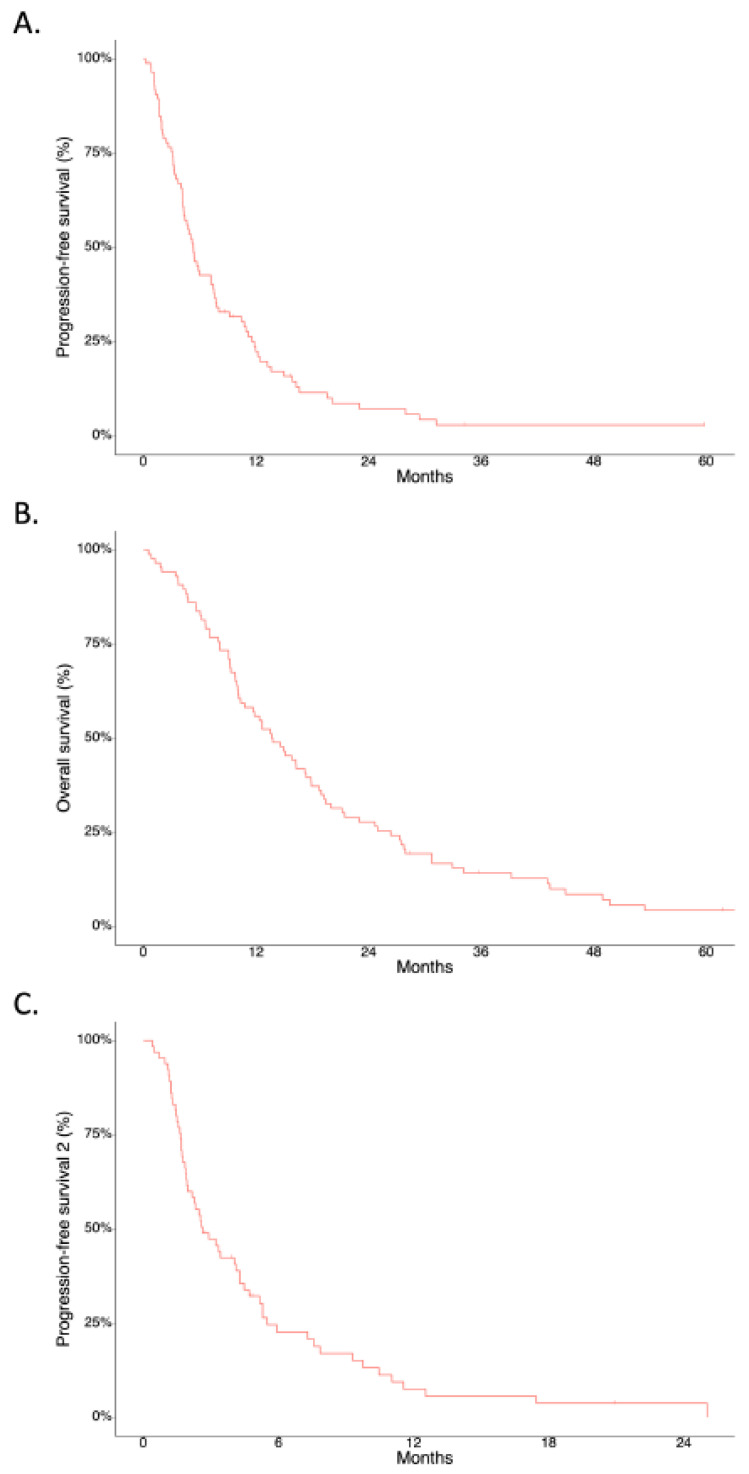
Survival outcomes of patients. (**A**) Progression-free survival after the first-line palliative chemotherapy (PFS1), (**B**) overall survival (OS), and (**C**) progression-free survival after the second-line palliative chemotherapy (PFS2).

**Figure 3 cancers-14-02383-f003:**
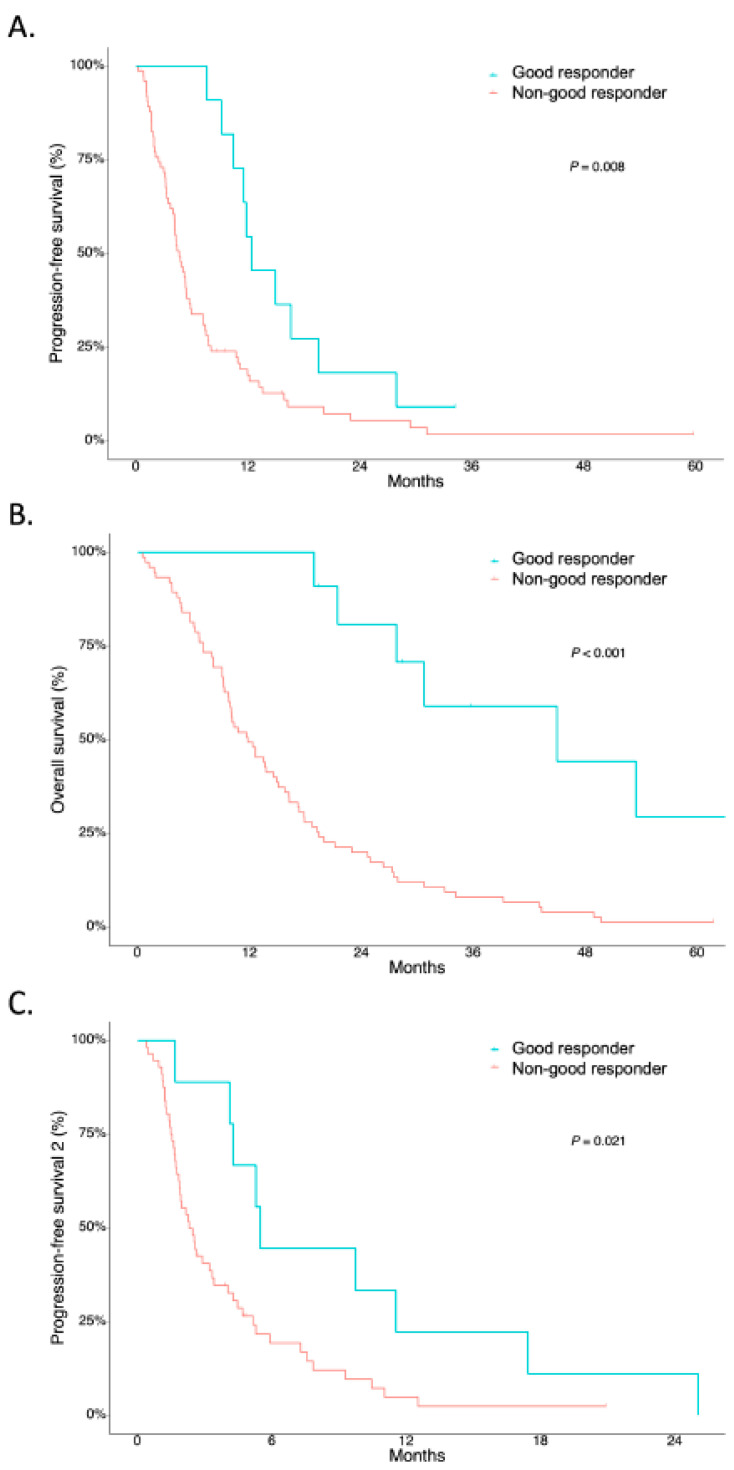
Survival outcomes according to patient groups: good responders and non-good responders. (**A**) PFS1, (**B**) OS, and (**C**) PFS2.

**Figure 4 cancers-14-02383-f004:**
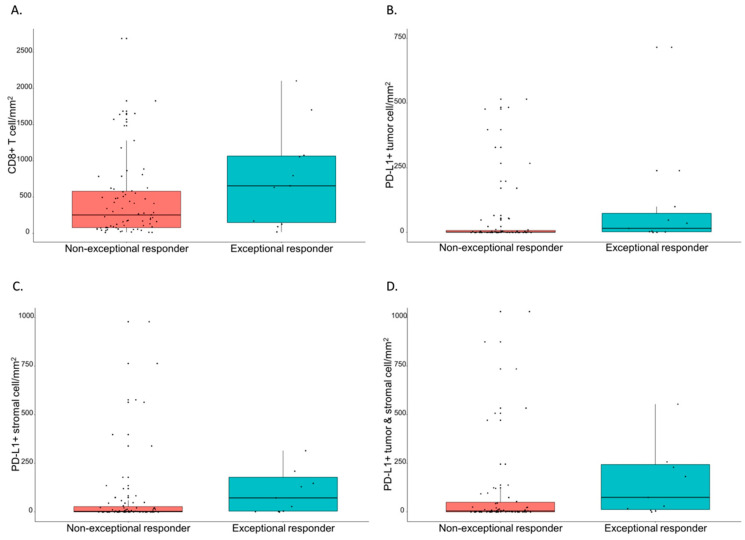
Immune profile according to patient groups: good responders and non-good responders. (**A**) CD8+ T cells, (**B**) PD-L1+ tumor cells, (**C**) PD-L1+ stromal cells, and (**D**) PD-L1+ tumor and stromal cells.

**Table 1 cancers-14-02383-t001:** Patient characteristics according to treatment response after palliative chemotherapy.

	Non-Good Responder (*n* = 75)	Good Responder (*n* = 11)	*p*-Value
Age, median (range)	60 (35–82)	66 (45–75)	0.321
Sex			>0.999
Male	44 (58.7%)	6 (54.5%)	
Female	31 (41.3%)	5 (45.5%)	
Primary site			0.613
Right colon	36 (48.0%)	4 (36.4%)	
Left colon	27 (36.0%)	4 (36.4%)	
Rectum	12 (16.0%)	3 (27.3%)	
Histologic group			0.814
Adenocarcinoma	69 (92.0%)	11 (100.0%)	
Mucinous	3 (4.0%)	0 (0.0%)	
Signet ring cell	1 (1.3%)	0 (0.0%)	
Others	2 (2.7%)	0 (0.0%)	
Tumor grade			0.725
Well differentiated	3 (4.0%)	1 (9.1%)	
Moderately differentiated	47 (62.7%)	7 (63.6%)	
Poorly differentiated	25 (33.3%)	3 (27.3%)	
Initial stage at diagnosis			0.770
I	1 (1.3%)	0 (0.0%)	
II	1 (1.3%)	0 (0.0%)	
III	15 (20.0%)	1 (9.1%)	
IV	58 (77.3%)	10 (90.9%)	
Disease setting			0.524
Initially metastatic	58 (77.3%)	10 (90.9%)	
Recurrent	17 (22.7%)	1 (9.1%)	
KRAS status			0.601
Wild	74 (98.7%)	10 (90.9%)	
Mutant	1 (1.3%)	1 (9.1%)	
NRAS status			NA
Wild	16 (100.0%)	4 (100.0%)	
Mutant	0 (0.0%)	0 (0.0%)	
Not evaluated	59	7	
MSI status			0.162
MSI	8 (14.3%)	1 (9.1%)	
MSS	48 (85.7%)	10 (90.9%)	
Not evaluated	19	0	
First-line palliative chemotherapy			0.233
FOLFIRI	16 (21.3%)	1 (9.1%)	
FOLFOX/XELOX	30 (40.0%)	3 (27.3%)	
FOLFIRI + Bevacizumab	9 (12.0%)	5 (45.5%)	
FOLFIRI + Cetuximab	3 (4.0%)	0 (0.0%)	
FOLFOX/XELOX + Bevacizumab	7 (9.3%)	1 (9.1%)	
Capecitabine	4 (5.3%)	1 (9.1%)	
Others *	3 (4.0%)	0 (0.0%)	
None **	3 (4.0%)	0 (0.0%)	

* Bevacizumab + XELOX + Simvastatin (2), Cetuximab + LGX818 + BYL719(1). ** Patients who could not receive palliative chemotherapy due to poor performance status and/or rapid progression of the disease.

**Table 2 cancers-14-02383-t002:** Immune profile of the BRAF-MT tumors in the non-good and good responder groups.

	Non-Good Responder (*n* = 75)	Good Responder (*n* = 11)	*p*-Value
CD8+ T cell/mm^2^, median (IQR)	254.29 (79.21–580.43)	656.00 (149.05–1067.14)	0.092
PD-L1+ tumor cell/mm^2^, median (IQR)	0.95 (0.00–7.32)	15.56 (2.13–73.81)	0.050
PD-L1+ stromal cell/mm^2^, median (IQR)	3.17 (0.00–27.30)	72.38 (3.65–178.25)	0.025
PD-L1+ tumor and stromal cell/mm^2^, median (IQR)	5.08 (0.36–49.68)	74.92 (12.38–243.49)	0.032
PD-1+ stromal cell/mm^2^, median (IQR)	45.08 (19.39–174.66)	325.40 (69.52–423.87)	0.046

## Data Availability

The datasets generated during and/or analyzed during the current study are available from the corresponding author on reasonable request.

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
