# Peer review of "Immune Profile of BRAF-Mutated Metastatic Colorectal Tumors with Good Prognosis after Palliative Chemotherapy"

_cancers, 2022, doi:10.3390/cancers14102383_

Round 1

Reviewer 1 Report

In this interesting original paper entitled “Immune profile of BRAF-mutated metastatic colorectal tumors with good prognosis after palliative chemotherapy”, Jeong Eun Kim et al. are presenting evidence that based on an immune profile analysis, a higher PD-L1 expression and CD8-positive cell infiltration were observed in BRAF-MT tumors with  a good prognosis in a cohort of unselected patients with BRAFV600E-MT CRC, similar to those in previous studies.

Among other studies, programmed cell death ligand-1 (PD- L1) expression has been shown to be associated with BRAF mutation. Also, in a recent retrospective analysis, BRAF mutation was suggested to be a potential predictive biomarker of immune checkpoint inhibitors in mismatch repair (MMR)-deficient CRC with a lower response rate and shorter PFS.

Therefore, the authors in the present study have evaluated the immune profile of the BRAFV600E.

MT CRC tumors and compared the differences in the immune profile of the tumors according to treatment response to investigate the association between immune microenvironment and prognosis of BRAF600E-MT CRC.

To this reviewer the data are being presented in a commendable systematically and didactic way.

The literature cited seems sufficient, adequate, relevant, up to date and appropriately interpretated. - Additionally, the article is well written.

Some remarks/suggestions for improval:

In this study BRAF mutation was found in 123 patients out of 2313, which amounts to 5,3%. This value is lower than the expected number of around 10% for BRAF-positive tumors in this tumor population. Thus, one might argue that the presented data and interpretation are not as typical for the CRC population as they should. This fact should be problematized in the discussion part.

However, as this reviewer sees it, the highly significant statistical difference in PFS1, OS, and PFS2 between the two groups (good responders vs. non-good responders) strongly indicates that the differences are biologically relevant.

In table 2, line 205-206. “Immune profile of the BRAF-MT tumors in the non-exceptional and exceptional responder.” Comment: The non-good responder is here changed to non-exceptional responder and the good responder is changed to exceptional responder. The designations of the two groups should be consistent throughout the text.

The text and numbers on the x- and y-axes of fig. 2 and 3 should be slightly increased. The text on the x- and y-axes in fig. 4 should be greatly increased. As it is now, it can hardly be read.

In fig. 4  a large spread of values is observed in the non-good responders, which naturally reduces the level of statistical significance between the two groups. In other words, there are quite many tumors having high values of (A) CD8+ T cells, (B) PD-L1+ tumor cells, (C) PD-L1+ stromal cells, and (D) PD-L1+ tumor and stromal cells in the non-good responders. This should be discussed.

The last sentence beginning, in line 89 …, “to palliative first-line chemotherapy of the study population.” is not clear to this reviewer: “GR was defined as patients with sustained partial response at least 6 months after the first- or second-line palliative chemotherapy, which was longer than the median time-to-progression, to palliative first-line chemotherapy of the study population.” Pls. rewrite.

Line 118. PD-L1 (1:500; CST) is not formally defined.

The Discussion part: This reviewer finds the discussion thorough, relevant and adequate. However, based on these findings, the potential clinical aspects/prospects based on the immunolandscape in BRAFV600E CRC should possibly be discussed in some more length.

Author Response

Thank you for your feedback and precise comments.

The answers to your questions are as follows.

Question) In this study BRAF mutation was found in 123 patients out of 2313, which amounts to 5,3%. This value is lower than the expected number of around 10% for BRAF-positive tumors in this tumor population. Thus, one might argue that the presented data and interpretation are not as typical for the CRC population as they should. This fact should be problematized in the discussion part.

Answer) Thank you for your comment. We discussed the relatively lower prevalence of BRAF mutation in our cohort. This might be explained by possible different prevalence according to race or ethnicity. The prevalence of BRAF mutation was reported as 5.6% in Asian patients in another study. Because of the retrospective nature of data collection of our study, the prevalence of BRAF mutation might be under-investigated with exception of patients without results of BRAF testing.

“ And the incidence of BRAF mutation was lower with 5.3% compared to previous reports. This might be explained by possible different prevalence according to race or ethnicity. The prevalence of BRAF mutation was reported as 5.6% in Asian patients in another study. Because of the retrospective nature of data collection of our study, the prevalence of BRAF mutation might be under investigated with exception of patients without results of BRAF testing.”

However, as this reviewer sees it, the highly significant statistical difference in PFS1, OS, and PFS2 between the two groups (good responders vs. non-good responders) strongly indicates that the differences are biologically relevant.

Question) In table 2, line 205-206. “Immune profile of the BRAF-MT tumors in the non-exceptional and exceptional responder.” Comment: The non-good responder is here changed to non-exceptional responder and the good responder is changed to exceptional responder. The designations of the two groups should be consistent throughout the text.

Answer) Thank you for your precise comment. We have checked again the designations of the two groups throughout the text. And changed the miswritten expression.

Question) The text and numbers on the x- and y-axes of fig. 2 and 3 should be slightly increased. The text on the x- and y-axes in fig. 4 should be greatly increased. As it is now, it can hardly be read.

Answer) Thank you for your precise comment. We have updated figures with a large size of text on the x- and y-axes.

Question) In fig. 4  a large spread of values is observed in the non-good responders, which naturally reduces the level of statistical significance between the two groups. In other words, there are quite many tumors having high values of (A) CD8+ T cells, (B) PD-L1+ tumor cells, (C) PD-L1+ stromal cells, and (D) PD-L1+ tumor and stromal cells in the non-good responders. This should be discussed.

Answer) Thank you for your comment. As you can see in table 2, there are also patients with a high number of CD8+ T cells, PD-L1+ tumor cells, PD-L1+ stromal cells, and PD-L1+ tumor and stromal cells in the non-good responder group and the value is higher than the median value of good responders in some patients. Considering the wide range of cell numbers of immune markers in both non-good and good responder groups, these outliers are inevitable in multiplex IHC analysis.  

Question) The last sentence beginning, in line 89 …, “to palliative first-line chemotherapy of the study population.” is not clear to this reviewer: “GR was defined as patients with sustained partial response at least 6 months after the first- or second-line palliative chemotherapy, which was longer than the median time-to-progression, to palliative first-line chemotherapy of the study population.” Pls. rewrite.

Answer) Thank you for your precise comment. We removed the phrase “or second line”. We have analyzed the response after the first line palliative chemotherapy.

Question) Line 118. PD-L1 (1:500; CST) is not formally defined.

Answer) We have updated the explanation as follows. programmed death-ligand 1 (PD-L1, 1:500; CST)

Question) The Discussion part: This reviewer finds the discussion thorough, relevant and adequate. However, based on these findings, the potential clinical aspects/prospects based on the immunolandscape in BRAFV600E CRC should possibly be discussed in some more length.

Answer) Thank you for your comments. We discussed the potential clinical aspects based on the immune microenvironment additionally in the discussion part.

“Even with the improvement of survival with the new treatment strategy in BRAF-MT CRC, there are still unmet needs because of the relatively short response duration of combination treatment of encorafenib and cetuximab. And the priority or sequence of this new treatment and immunotherapy is still under debate, especially in MSI BRAF-MT CRC. With our study, we can suggest a clue to a combination of immunotherapy with the selection of patients based on the immune profile of BRAF-MT tumors. The immune profile of BRAF-MT CRC can be evaluated in future clinical trials or in the prospective cohort to enhance the understating of tumor biology of BRAF-MT tumors and find a new treatment strategy for BRAF-MT CRC.”

Reviewer 2 Report

Dear Authors

In the article, there are many chemotherapy formulas, which are not uniform. The immune profiles listed in Table 2 are greater in the GR group. But the result cannot explain the immune profiles that could make so many chemotherapy formulas work well. How can the authors explain the result? The result may be found in BRAF WT CRC with same immune profiles.

Author Response

Question) In the article, there are many chemotherapy formulas, which are not uniform. The immune profiles listed in Table 2 are greater in the GR group. But the result cannot explain the immune profiles that could make so many chemotherapy formulas work well. How can the authors explain the result? The result may be found in BRAF WT CRC with same immune profiles.

Answer) Thank you for your comments. As you pointed out, we included a heterogeneous patient population who received many kinds of chemotherapy. However, the tumor tissue used for the multiplex IHC to evaluate the tumor immune profile was obtained before treatment. Although we could not clearly explain the association between immune profile and the effect of chemotherapy with this small size study, we can suggest a clue that de novo difference in the immune profile of BRAF-MT CRC might be associated with treatment responses. But as you mentioned, these analyses should be also conducted in patients with BRAF WT CRC to explain that these features in tumor biology are confined to BRAF MT tumors. We additionally discuss this limitation in the discussion part.

“Furthermore, there is a possibility that we cannot clearly explain the association between immune profile and the effect of chemotherapy considering heterogeneous chemotherapy regimens. But we can suggest a clue that the de novo difference in the immune profile of BRAF-MT CRC might be associated with treatment responses considering that the tumor tissue used for the multiplex IHC to evaluate the tumor immune profile was obtained before treatment. However, these analyses should be also conducted in patients with BRAF WT CRC to explain that these features in tumor biology are confined to BRAF MT tumors. We additionally discuss this limitation in the discussion part."

Reviewer 3 Report

Here are a few points for this manuscript:

  1. Why were only 4 biomarkers quantified? Please describe how you chose the 4 biomarkers, but not others.
  2. Why were the 3 patients who did not receive palliative chemotherapy included in the analysis? The study design and selection of patients should be consistent.
  3. Figure 2. It’s better to plot the 3 survival curves in the same figure.
  4. Cancer cell exhibits immune escape by the expression of PD-L1. But this study observed higher expression of PD-L1 in both tumor and stromal cells in the responding group. While the authors discussed this observation and suggested that tumor heterogeneity, patient populations, and complex immune microenvironment could explain such contrasting patterns, it is important for the authors to either include additional independent datasets to validate their findings or provide a more detailed and comprehensive discussion of the advantageous effect of high PD-L1 expression of cancer cells in increasing treatment response.

Author Response

Q1) Why were only 4 biomarkers quantified? Please describe how you chose the 4 biomarkers, but not others.

A1) Thank you for your comment. To evaluate the immune profile of BRAF-MT tumors, 4 markers might not sufficient. There was a limitation in the number of markers for multiplex IHC analysis at the same time when this study was performed at our center because it was the beginning project to set up multiplex IHC studies. We discussed with the specialist who is in charge of staining and analysis of multiplex IHC and selected well-known representative immune markers. We discussed this additionally as a limitation of our study.

“In addition, only 4 markers were selected to evaluate the immune profile of BRAF-MT tumors due to a limitation in the number of markers for multiple IHC analysis at that time when this study was performed at our center. It was not sufficient to comprehensively assess the immune profile, but we cautiously selected well-known representative immune markers.”

Q2) Why were the 3 patients who did not receive palliative chemotherapy included in the analysis? The study design and selection of patients should be consistent.

A2) Thank you for your comment. Three patients who did not receive palliative chemotherapy in the non-good response group could not receive palliative chemotherapy due to poor performance status and/or rapid progression with lab abnormalities just before the planned infusion of chemotherapy. For comparison of the immune microenvironment of BRAF tumors, these patients’ tissues were included in the non-good response groups because of the small number of samples and the accordance between response after chemotherapy and survival in our patient group.

Q3) Figure 2. It’s better to plot the 3 survival curves in the same figure.

A3)Thank you for your comments. To prevent confusion in PFS1 and PFS2, we plotted 3 survival curves separately.

Q4) Cancer cell exhibits immune escape by the expression of PD-L1. But this study observed higher expression of PD-L1 in both tumor and stromal cells in the responding group. While the authors discussed this observation and suggested that tumor heterogeneity, patient populations, and complex immune microenvironment could explain such contrasting patterns, it is important for the authors to either include additional independent datasets to validate their findings or provide a more detailed and comprehensive discussion of the advantageous effect of high PD-L1 expression of cancer cells in increasing treatment response.

A4) Thank you for your comment. As you mentioned, the finding of higher expression of PD-L1 in the good response group is contrary to previously reported studies that suggested immune escape by the expression of PD-L1 in cancer cells. However, the prognostic value of PD-L1 expression and the role of PD-L1 in the immune microenvironment is not clear with conflicting results. As you pointed out our study is not sufficient to prove the definite effect of high PD-L1 expression on treatment response without the same results in the validation cohort. But we can suggest that the immune profile of BRAF-MT tumors might be one of the important biologic features related to clinical outcomes.

Round 2

Reviewer 2 Report

Dear Authors

 I think the answer is appropriate. Hope you have follow-up studies on BRAF-WT patients with this issue.